# A Worldwide View on the Reachability of Encrypted DNS Services

## ABSTRACT

To protect user DNS privacy, DNS over TLS (DoT), DNS over HTTPS (DoH), DNS over QUIC (DoQ), and DNS over HTTP/3 (DoH3) are proposed to encrypt DNS traffic. Collectively, we term them DNS over Encryption (DoE). Existing studies have preliminarily measured the reachability of DoE services. However, they either focus on a few DoT/DoH domains or a few vantage points (VPs).

In this paper, we present the first comprehensive worldwide view of DoE service reachability. By collecting data from our 15-month-long scan, we elaborately built a list of 1302 operational DoE domains as measurement targets, 448 of which support IPv6. Then we performed 10M DoE over IPv4 (DoEv4) and 570K DoE over IPv6 (DoEv6) queries from 5K VPs over a two-month period, encompassing 102 countries. Our results reveal that the accessibility of DoE services is poor in some regions. Specifically, 592K DoEv4 queries and 28K DoEv6 queries were blocked during our measurements. Internet not free countries more often block DoEv4 queries by interfering with TCP connections and QUIC version negotiation. Compared to DoEv4, the reachability of DoEv6 services is better. In particular, some DoE blocking policies target only specific IP addresses or DoE protocols, providing clients with the opportunity to access blocked DoE domains. Our study highlights the need for the community to pay attention and improve the reachability of DoE services.

## 1 INTRODUCTION

Domain Name System (DNS) was initially designed based on UDP or TCP protocols, which lacks privacy and security protection [6, 37]. One promising mitigation approach is to encrypt DNS traffic. To this end, four encrypted DNS protocols, DNS over TLS (DoT) [26], DNS over HTTPS (DoH) [24], DNS over QUIC (DoQ) [27], and DNS over HTTP/3 (DoH3) [27], were standardized by the IETF community. In this paper, we term them collectively as DNS over Encryption (DoE). Currently, many DNS providers [3, 39, 56], clients [8, 38, 44], and operating systems [22, 30, 52] already support DoE.

Since DoE may be abused by malicious attackers [48], and users may utilize DoE to bypass DNS regulation [10]. Some countries/ISPs have blocked DoE queries to maintain their grip on Internet governance [4]. However, previous work only preliminarily measured the reachability of DoE services [5, 23, 35], either by focusing on a few DoT/DoH domains or a few vantage points (VPs). Furthermore, no studies comprehensively evaluated the blocking types of DoE services and the connectivity of DoE services over IPv6.

**Challenges.**   Considering the dependence of other protocols on DoE [21, 50] and QUIC censorship in some countries [16], it is imperative to thoroughly assess the reachability of DoE service. This task primarily encounters the following two challenges.

Firstly, many DoE servers are unable to reliably serve users [34, 35], and the community lacks a public comprehensive list of DoE domains. Therefore, it is necessary to meticulously collect operational DoE servers. Secondly, blocking behaviors may occur at various stages of DoE communication. Therefore, it is crucial to systematically monitor all levels of the network stack from the global VPs, and identify different types of DoE blocking.

**Our study.**   In this paper, we measure the reachability of DoE services through the following three steps.

Firstly, we conducted a 15-month-long Internet-wide scan to collect open DoE resolvers and implement an automated method to filter operational DoE servers. Ultimately, we obtained 1302 DoE over IPv4 (DoEv4) and 448 DoE over IPv6 (DoEv6) domains.

Secondly, we collected 5031 and 473 VPs that support sending DoE queries to IPv4 and IPv6 addresses, respectively. In the end, we performed 10M DoEv4 and 560K DoEv6 queries from 102 countries over the course of two months.

Thirdly, we monitored the entire process of DoE communication, encompassing the resolution of DoE domains to the reception of DoE responses from global VPs. At last, we observed seven blocking types, including Pre-resolve, Ping, TCP, TLS, QUIC version negotiation, QUIC, and Response blocking.

Based on our measurement results, we can answer the following research questions: *How is the global deployment of DoE services* (see Section 4.1)? *Which regions block DoEv4 services* (see Section 4.2)? *What blocking types do DoEv4 services suffer* (see Section 4.3)? *How is the reachability of DoEv6 service* (see Section 4.4)? *How many DoE queries are censored* (see Section 4.5)? *Can clients access blocked DoE domains* (see Section 4.6)?

**Major findings.**   Throughout 15 monthly scans, the number of open DoT/DoH IPv4 addresses consistently remained at 20K/11K. In addition, the number of open DoQ/DoH3 IPv4 addresses increased significantly, eventually stabilizing at 3.7K/300. However, the number of stable DoE domains, which provide DoE services for three consecutive months, is only about 1K and generally remains steady.

Considering DoE service reachability, our results reveal that 592K DoEv4 queries and 28K DoEv6 queries are blocked. VPs located in China, Indonesia and Vietnam exhibit the worst reachability to DoEv4 services. In addition, some autonomous systems (ASes) in Russia and Ukraine obviously block DoH3 services. The majority of DoEv4 service blocking occurs when VPs ping DoE servers, and about one-third of them are VPs try to connect DoE servers in China. Furthermore, certain VPs are unable to obtain authentic IP addresses of DoE domains. In particular, the reachability of DoEv4 services is poor in Internet not free countries, and they are more often blocked during the TCP connection and QUIC version negotiation. We also observe behavior strongly indicative of censorship in 27.18% of blocked DoEv4 queries and 19.73% of blocked DoEv6 queries.

The reachability of DoEv6 services is generally better, especially for DoQv6 and DoH3v6. The TLS handshake failure is the primary cause of DoEv6 service unreachability. Furthermore, our results suggest that many DoE service blocking policies are defective, as they allow clients to access blocked DoE domains by changing IP addresses or DoE protocols. For example, 96/120 blocked DoTv4 domains can provide DoTv6/DoHv4 services in China.

We hope our study can drive future efforts to improve the reachability of DoE services. To help the community reproduce and

extend our research, we publish our code and collected data at https://github.com/DNS-over-Encryption/Reachability/.

## 2 BACKGROUND AND RELATED WORK

In this section, we first outline DoE protocols. Then, we present previous research related to our work.

### 2.1 DNS over Encryption protocols

In recent years, encrypted DNS has emerged as one of the consensus approaches to mitigate active manipulation and passive monitoring of DNS traffic [3, 39, 56]. We present the comparison of four DoE protocols in Table 1.

**Table 1: Comparison of four DoE protocols.**

| DoE | Port | Underlying protocol | Server template |
|-----|------|---------------------|-----------------|
| DoT | TCP/853 | TCP+TLS | dns.nextdns.io |
| DoH | TCP/443 | TCP+TLS+HTTP | https://dns.nextdns.io/dns-query |
| DoQ | UDP/853 | UDP+QUIC | dns.nextdns.io |
| DoH3 | UDP/443 | UDP+QUIC+HTTP | https://dns.nextdns.io/dns-query |

Standardized in 2016 [26] and 2018 [24], DoT and DoH utilize TLS sessions to encrypt DNS packets and embed DNS queries into TCP and HTTP messages, respectively. However, the performance of DoT and DoH suffers from the unavoidable overhead introduced by TCP and TLS. Two QUIC-based DNS protocols, DoQ and DoH3, were introduced in 2022 [27] to protect user DNS privacy. Benefiting from the advantages of QUIC, DoQ/DoH3 can provide security properties similar to DoT/DoH while improving performance.

The client relies on URI templates to locate DoH/DoH3 services and sends DoH/DoH3 requests using the GET or POST method. Unfortunately, the IETF has not defined a standard path template for DoH/DoH3. In addition, since DoT and DoQ run on a dedicated port 853, attackers or firewalls can easily identify and block their traffic. Considering the community's preference for DoH [8, 30, 38], DoH3 may get better support in the future, which is confirmed by Google's announcement of adding DoH3 support in Android [22].

### 2.2 Related work

DNS manipulation is widespread in the wild, causing numerous security and privacy concerns [43]. Several works have evaluated the efficacy of encrypted DNS protocols in circumventing DNS manipulation. Specifically, Jin et al. [31] revealed that 37% of censored domains are accessible in China by using DoT/DoH. Moreover, Hoang et al. [23] reported that DoH and ESNI [50] enable over half of the censored domains to evade blocking in China.

To utilize DoE for preventing DNS manipulation, the client first needs to ensure that the DoE server is accessible. However, previous studies only preliminarily evaluated the accessibility of DoT and DoH services over IPv4 in the wild. In 2019, Lu et al. [35] measured the reachability of three public DoT/DoH servers. They pointed out that the reachability of DoT/DoH service is affected by censorship and TLS interception. In addition, Basso et al. [5] analyzed the blocking of 123 DoT/DoH servers in Kazakhstan, Iran, and China. They found that 50% of DoT servers are blocked in Iran, and Cloudflare/Google services are highly censored. After that, Hoang et al. [23] evaluated the accessibility of 12 DoT and 59 DoH servers in

85 countries. Their results disclosed strict censorship of DoT/DoH services in China, Russia, Iran, Saudi Arabia, and Venezuela. Furthermore, Jin et al. [31] investigated DNS manipulation on 3818 DoT and 75 DoH IP addresses. They discovered that more than two-thirds of DoT/DoH services manipulate DNS responses from at least one domain. Regrettably, the community currently lacks comprehensive awareness of the reachability of global DoE services, which is our dedicated work to focus on.

## 3 METHODOLOGY

In this section, we first introduce our collection process of DoE domains. Then, we describe our method of DoE reachability measurement. Figure 1 illustrates the workflow of our research methodology. Finally, we discuss the ethics and limitations of our study.

### 3.1 DoE domain collection

Our study aims to evaluate the global reachability of DoE services. However, numerous open DoE servers are merely artifacts of some providers that do not serve real-world users [34, 35]. It is not appropriate to include these servers in our evaluation.

Therefore, the first problem we should solve is the lack of a comprehensive list of operational DoE servers. Specifically, operational DoE servers are expected to meet the following criteria: 1) replying correct DoE responses; 2) holding usable domain names; 3) configuring valid certificates; 4) providing continuous DoE service. To this end, we first perform long-term scans to discover open DoE domains, and then select operational DoE domains.

**DoE domain discovery.** We discover open DoE domains through the following three steps. The first step is to scan IPv4 addresses that open DoE ports. The second step is to identify IPv4 addresses that can correctly respond to DoE queries. The third step is to associate IP addresses with DoE domains.

1) *Scan port.* Our study only considers DoE services deployed on standard ports. In practice, we use ZMap [15] to obtain all IPv4 addresses opening ports TCP/853, TCP/443, UDP/853, and UDP/443.

2) *Identify service.* Identifying open DoT/DoQ servers is simple. If the IP address correctly responds to DoT and DoQ requests on port TCP/853 and port UDP/853, we consider it as a DoT and DoQ server, respectively. All DoE requests only lookup the A record of our domain name, which is hosted on our authoritative nameservers.

Since the lack of a standard URI template, identifying open DoH/DoH3 servers is relatively complicated. To find as many servers as possible, we first need to determine common path templates. Based on some public lists [12, 47] and previous studies [34–36], we select four path templates (/dns-query, /query, /resolve, and /) to construct URI templates. After that, we probe each IP address opening port TCP/443 with 16 test suites, which comprise four path templates, two HTTP methods (GET, POST), and two HTTP versions (HTTP/1.1, HTTP/2). Furthermore, since DoH3 servers only support HTTP/3, 8 of the 16 test suites are applied to IP addresses opening port UDP/443. If any test suite successfully responds to our DoH and DoH3 requests, we consider the corresponding IP address provides DoH and DoH3 services, respectively.

3) *Associated domain name.* The CN value in the subject field and DNS names in the SAN extension list all domains protected by the certificate [11]. Therefore, we can extract domains associated

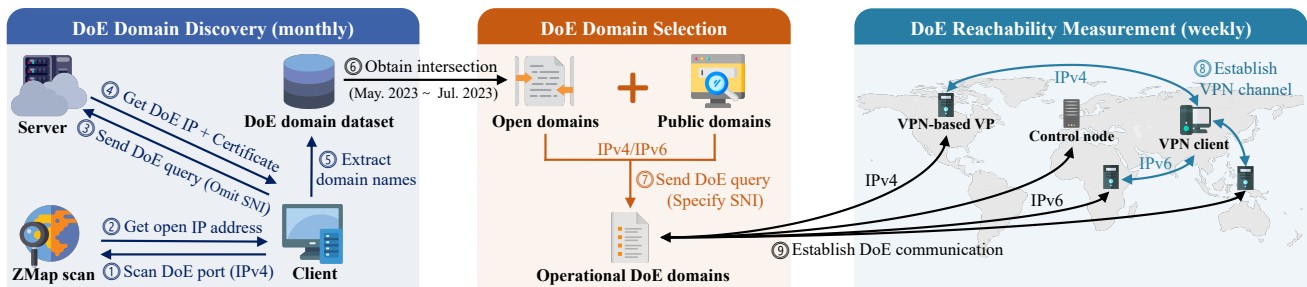

Figure 1: Workflow for DoE domain collection and reachability measurement of DoE services.

with DoE IP addresses through leaf certificates saved during DoE service identification. However, not all domain names listed in the certificate are used for DoE services. Based on previous reports [20, 34, 47], we only retain non-wildcard domain names that include "dns", "dot", "doh" or "doq". At last, we build DoE domain datasets.

From July 2022 to September 2023, we monthly repeated the above scanning process from Hong Kong.

**DoE domain selection.** The DoE protocol is currently designed specifically for client usage [24, 26, 27]. As such, only reachability measurements for operational DoE servers are meaningful.

Recall the four criteria of operational DoE servers. One of them is providing continuous DoE service. To this end, we obtain the intersection of DoE domain datasets collected from May 2023 to July 2023 as a candidate list, and supplement it with some public DoE domains [12, 47]. To satisfy the other three criteria, we first resolve the IPv4/IPv6 addresses of candidate DoE domains and apply the method mentioned above for identifying DoE services to these IP addresses. Remarkably, we specify the DoE domain as the value of the SNI field during the TLS and QUIC handshake. Then, we reserve DoE domains for which all IP addresses respond correctly and configure valid certificates as targets for our reachability measurement. In particular, we refer to the remaining 1302/448 operational DoE servers that support IPv4/IPv6 as DoEv4/DoEv6 domains.

## 3.2 Reachability measurement

Service unreachability may arise from deliberate behavior of network middleware or target servers. For instance, ISPs can restrict local users from using DoE services [23], and DoE servers can deny access from unauthorized users [31]. Hence, our measurement platform needs globally distributed VPs capable of monitoring the entire DoE communication process.

**Vantage points.** To avoid ethical issues arising from human participation, we collect VPN-based VPs from eight commercial VPN providers. Due to the lack of stable VPN servers in the Chinese mainland, we deploy two EC2 cloud instances located in Beijing, China and Hangzhou, China.

Considering commercial platforms may falsely claim server locations, we use ip-api [28] to verify the geolocation of each VP. Furthermore, providers may implement DNS hijacking on their servers, which affects the resolution of DoE domains. As such, we lookup the A record of our domain name[1] from all VPs to two

popular DNS providers (8.8.8.8, 1.1.1.1). Subsequently, we examine whether the DNS resolver querying our authoritative nameservers belong to two popular DNS providers [1, 2]. Our investigation uncovers DNS query hijacking by the NordVPN [42], affecting queries directed to 8.8.8.8, and by the Surfshark [54], affecting both 8.8.8.8 and 1.1.1.1. Ultimately, we removed 324 unreliable VPN nodes.

Given the potential occurrence of server downtime and spurious responses, determining that the DoE service is blocked relies on the comparison of measurement results from VPs and control nodes. To this end, we deploy five EC2 cloud instances in Hong Kong, Frankfurt, Virginia, São Paulo, and Sydney as our control nodes.

**Blocking types.** Accurate classification of blocking types is pivotal for evaluating service reachability. According to the DoE query process shown in Figure 2, we define seven blocking types and describe threat models[2] concerning network middleware (e.g., firewalls, censors, and ISPs).

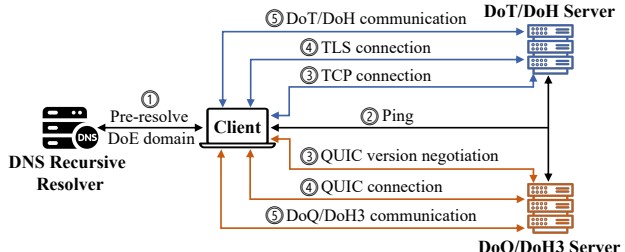

Figure 2: Process of our VP accessing the DoE domain.

1) *Pre-resolve blocking*: The client is unable to obtain authentic IP addresses of the DoE domain through DNS lookup. Since the DNS query is in plain text, network middleware can easily intercept these lookups and return either empty responses or forged IP addresses. In such cases, the client is unable to establish subsequent DoE connections, or they may face redirection to a fake server.

2) *Ping blocking*: The client is unable to receive ICMP packet responses from the DoE server. Network middleware can completely prevent clients from connecting to DoE servers based on the IP address. This is the most direct way to implement blocking policies, but it often results in extensive collateral damage.

3) *TCP blocking*: The client is unable to establish a TCP connection with the DoT/DoH server on port TCP 853/443. Network

---

[1]The domain name in each lookup includes a unique random string. This ensures that our authoritative nameservers can receive queries from DNS resolvers.

[2]The implementation of service blocking by target servers, according to their security policies or service scopes, is relatively straightforward.

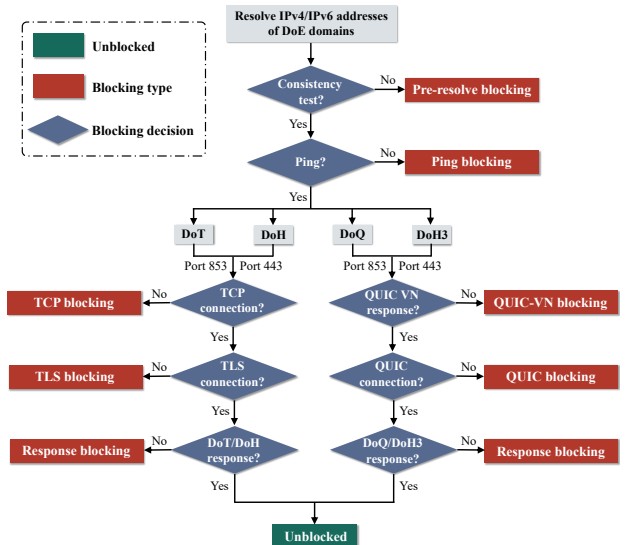

**Figure 3: Flowchart of detecting DoE service blocking.**

middleware can inspect TCP packet headers and port numbers to intercept TCP traffic for specific IP addresses.

4) *TLS blocking*: The client is unable to complete a TLS handshake with the DoT/DoH server on port TCP 853/443. The TLS handshake exposes many sensitive information, such as the server domain name, server certificate, and ALPN. As such, network middleware can implement complex blocking strategies to block the TLS connections or return invalid server certificates to clients.

5) *QUIC-VN blocking*: The client is unable to complete a QUIC version negotiation (QUIC-VN) with the DoQ/DoH3 server on port UDP 853/443. This indicates that network middleware directly intercept the QUIC session between the client and the DoE server, without considering contents in the subsequent QUIC traffic.

6) *QUIC blocking*: The client is unable to establish a QUIC connection with the DoQ/DoH3 server on port UDP 853/443. Certain sensitive information, such as the server domain name and ALPN, is exposed in the initial packet during QUIC handshake. Network middleware can intercept or refuse QUIC connections.

7) *Response blocking*: The client is unable to receive correct DoE responses from the server. Once the TLS/QUIC encrypted channel is established, network middleware between the client and the DoE server can only intercept the DoE session without the capacity to modify its contents[3]. However, clients might receive inaccurate DNS results. This may arise from the manipulation of the DNS session between the DoE server and the authoritative server, or from the authoritative server replying with incorrect IP addresses.
**Blocking detection.**    Our control nodes are responsible for connecting to VPN-based VPs and detecting blocking behavior. Figure 3 presents the flowchart for the detection of DoE service blocking.

At first, the VP uses Google DNS (8.8.8.8) to resolve the IPv4/IPv6 addresses of the tested DoE domain (test.doe.com). If the VP receives the DNS error code (e.g., REFUSED), empty DNS response,

bogon IP address [29], or timeout error, we consider that *Pre-resolve blocking* occurs. Otherwise, we perform a consistency test on each tested IP address (t.e.s.t) to determine whether it is forged. The consistency test involves three ground truths as follows.

1) $GT_{as}$: We resolve IP addresses of test.doe.com from five control nodes and use ip-api [28] to obtain the AS for all IP addresses. Then, we aggregate all AS results as the $GT_{as}$ for test.doe.com.

2) $GT_{title}$: We send HTTP GET requests to https://test.doe.com/ from five control nodes. Then, we aggregate all <title> tags in the page contents as the $GT_{title}$ for test.doe.com.

3) $GT_{prompt}$: Many DoE servers return user-friendly prompts for malformed DoE requests[4]. As such, we send HTTP GET requests to https://test.doe.com/dns-query from five control nodes. Then, we aggregate all prompts as the $GT_{prompt}$ for test.doe.com.

The detailed process of the consistency test is as follows.

1) We check whether the AS of t.e.s.t is in the $GT_{as}$. If yes, we consider t.e.s.t is authentic. Otherwise,

2) We try to establish a TLS connection from the control node with t.e.s.t and include test.doe.com in the SNI extension. If the TLS connection fails, we consider t.e.s.t is forged. Otherwise,

3) We verify whether server certificates received in the TLS handshake is valid. If yes, we consider t.e.s.t is authentic. Otherwise,

4) We send an GET request to https://test.doe.com/ from the control node. If the response contains the <title> tag in the $GT_{title}$, we consider t.e.s.t is authentic. Otherwise,

5) We send an GET request to https://test.doe.com/dns-query from the control node. If the response contains the prompt in the $GT_{prompt}$, we consider t.e.s.t is authentic, and vice versa.

If all control nodes determine that the tested IP is forged, we consider that *Pre-resolve blocking* occurs. Otherwise, we then ping the tested IP address. If the VP fails to receive correct ICMP responses, we consider that *Ping blocking* occurs. Otherwise, we establish subsequent connections with the tested domain, determined by the type of DoE service it supports.

Regarding DoT/DoH, the VP tries to establish a TCP connection with the tested IP address on port TCP 853/443. If it fails, we consider *TCP blocking* occurs. Otherwise, the VP tries to establish a TLS connection with the tested IP address. Regarding DoQ/DoH3, the VP tries to perform QUIC version negotiation with the tested IP address on port UDP 853/443. If the VP does not receive a valid QUIC version, we consider *QUIC-VN blocking* occurs. Otherwise, we establish a QUIC connection with the tested IP address.

During the TLS and QUIC handshake, we specify the SNI field as the tested DoE domain. If the TLS/QUIC connection establishment fails or the DoE server certificate is invalid, we consider *TLS blocking*/*QUIC blocking* occurs. Otherwise, the VP sends a DoE request to the tested IP address to lookup the A record for our domain name. If the VP does not receive a correct DNS response, we consider *Response blocking* occurs. Otherwise, we consider the DoE query is *Unblocked*.

Particularly, we detect each tested DoE domain three times. We only consider the DoE domain blocked if blocking occurs in all three detections. Since the differences in the blocking types suffered by blocked DoE domains in the three detections are minimal, this paper

---

[3]Network middleware that hold cryptographic keys or valid certificates of DoE servers can modify the content of the DoE response.

[4]For example, dns.google returns "Your client has issued a malformed or illegal request. Query must have a valid 'dns' parameter".

focuses only on the last blocked query. Furthermore, we perform daily scans of DoE domains from control nodes and remove all inaccessible domains. From August 7, 2023 to October 9, 2023, we weekly repeated the above reachability measurement.

### 3.3 Ethics and limitations

Since our study involves large-scale network scanning, we have the following ethical considerations. We scan for open ports and DoE services on a monthly basis, and close connections immediately after completing service identification. We set up reverse DNS records for our scanning platforms and provide measurement details on the corresponding websites. We did not receive any opt-out requests during our scan. Since human participation in reachability testing inevitably raises ethical issues, all of our VPs are commercial VPN nodes or cloud servers, and we only resolve our domains through DoE servers. Furthermore, we rate-limit requests sent by VPs to minimize traffic burden and measurement errors. Overall, the risks posed by our measurements are limited and controllable. Compared with the pressure on Internet infrastructure, we believe that our research can bring more benefits to communities and users.

Regrettably, VPN nodes are typically located in commercial data centers and Internet free areas, which means we can only obtain the lower bound of DoE service blocking. In addition, our method cannot accurately distinguish whether the blocking is caused by DoE servers or network middleware.

## 4 RESULTS

In this section, we first introduce our DoE server dataset and VP distribution. Then, we evaluate the reachability of DoEv4 and DoEv6 services. Following this, we analyze whether DoE services are blocked due to censorship. Finally, we investigate the incomplete blocking of DoE domains.

### 4.1 Dataset

**Open DoE servers.** Figure 4 shows the number of open DoE servers per scan over a 15-month period. The histogram represents the number of DoE domains, aligning with the left y-axis. The broken line represents the number of DoE IPv4 addresses, aligning with the right y-axis. In particular, we define a server that provide DoE services for three consecutive months as a stable DoE IP address/domain.

Since July 2022, we observe a relative stability in the number of DoT/DoH IPv4 addresses, while the number of DoQ/DoH3 IPv4 addresses is on the rise overall. Furthermore, Kosek et al. [32] found only 1217 open DoQ IPv4 addresses in April 2022. The above trends are mainly due to the fact that DoQ/DoH3 is standardized by RFC 9250 [27] in May 2022. In addition, the number of stable DoE domains is consistently steady, whereas the number of stable DoE IPv4 addresses exhibits fluctuations. This also illustrates the importance of our meticulous selection of operational DoE domains.

Furthermore, the number of DoE domains is significantly smaller than DoE IPv4 addresses. Digging deeper, we find two main reasons. Firstly, many DoE servers are embedded in firewalls or proxies that are not designed to offer usable domains for real-world users. For example, in May 2023, we observed 3896 DoT servers belonging to FortiGate [19] firewalls that were configured with self-signed

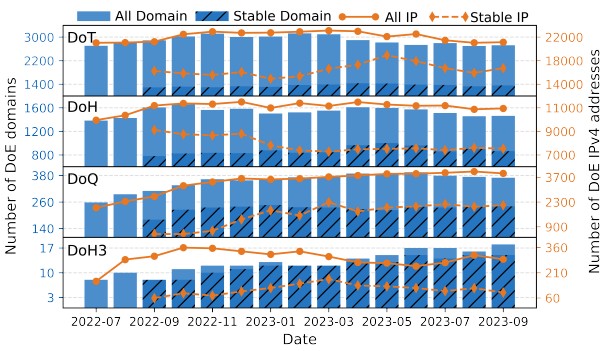

**Figure 4: Number of open DoE servers per month.**

certificates. These certificates only contained domains following the GT[.*] format. Secondly, some organizations configure the same certificate for their DoE servers. For example, in May 2023, we observed that certificates for 2491 DoQ IPv4 addresses, belonging to the NextDNS [40], were associated only with dns.nextdns.io.

**Operational DoE servers.** As illustrated in Table 2, we collect 1302 DoEv4 and 448 DoEv6 domains, most of which are located in Germany, the United States and China. Furthermore, about 95% of DoH/DoH3 domains support the /dns-query path template. To the best of our knowledge, our DoE domain dataset is the most comprehensive one to date. Specifically, [5, 13, 14, 20, 31, 35] only identified DoT/DoH IPv4 addresses; [23, 34, 36] only gathered DoT/DoH domains; and [32, 33] only collected DoQ IPv4 addresses.

**Vantage points.** As indicated in Table 3, we collect 5031 VPs, 473 of which support IPv6. According to Internet Freedom Scores [25], our VPs cover 15 of 21 not free (NF) countries and 17 of 32 partially free (PF) countries. Compared with other studies that use VPN nodes to measure the DoE service reachability, our VPs cover the

**Table 2: Number and top-3 countries of DoE domains.**

| Operational DoEv4 domains (1302) | | | |
|---|---|---|---|
| **DoT** | 1143 | **DoH** | 565 (716)[1] |
| **Country** | | **Country** | |
| Germany | 279 (24.41%) | United States | 114 (20.18%) |
| United States | 173 (15.14%) | Germany | 80 (14.16%) |
| China | 89 (7.79%) | China | 36 (6.37%) |
| **DoQ** | 240 | **DoH3** | 15 (24)[1] |
| **Country** | | **Country** | |
| China | 32 (13.33%) | United States | 4 (26.67%) |
| Germany | 32 (13.33%) | Cyprus | 3 (20.00%) |
| United States | 30 (12.50%) | Australia | 2 (13.33%) |
| Operational DoEv6 domains (448) | | | |
| **DoT** | 400 | **DoH** | 180 (234)[1] |
| **Country** | | **Country** | |
| Germany | 124 (31.00%) | Germany | 38 (21.11%) |
| United States | 61 (15.25%) | United States | 35 (19.44%) |
| France | 31 (7.75%) | Denmark | 21 (11.67%) |
| **DoQ** | 38 | **DoH3** | 13 (19)[1] |
| **Country** | | **Country** | |
| France | 5 (13.16%) | United States | 4 (30.77%) |
| United States | 4 (10.53%) | Cyprus | 3 (23.08%) |
| Japan | 3 (7.89%) | Australia | 2 (15.38%) |

[1] In parentheses is the number of URIs of DoH/DoH3 domains.

most countries (e.g., 102 in our work vs. 85 in [23]). However, our VPs in eight countries only cover one AS, which may bias the assessment for these countries.

**Table 3: Geographic distribution of vantage points.**

|         | IPv4 | IPv6 |           | IPv4  | IPv6  |
|---------|------|------|-----------|-------|-------|
| **VP**  | 5031 | 473  | **Continent** |   |   |
| **AS**  | 105  | 35   | Asia      | 33/48 | 14/48 |
| **Country** |  |      | Africa    | 5/54  | 1/54  |
| Total   | 102  | 42   | N. America | 9/23 | 2/23  |
| NF[1]   | 15   | 5    | S. America | 10/12 | 2/12 |
| PF[1]   | 17   | 5    | Europe    | 42/44 | 31/44 |
| F[1]    | 72   | 32   | Oceania   | 3/14  | 2/14  |

## 4.2 Which regions block DoEv4 services?

During our measurement period, we sent 10M DoEv4 queries to 1302 DoEv4 domains from 5K VPs, of which 592K queries were blocked. Our results show that in nine countries, the blocked ratio of DoEv4 queries performed by VPs is higher than 10%. Figure 5 plots the blocked ratio of DoEv4 queries performed by VPs in each country. We can observe that DoEv4 queries in China[5] are extremely blocked (36.11%), which is consistent with the findings of previous studies [4, 23]. Furthermore, since Russia and Ukraine implement censorship of HTTP/3 traffic [17], VPs located in them also exhibit obvious blocking of DoH3v4 services.

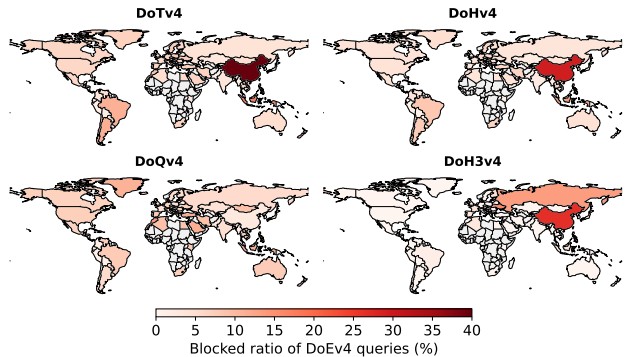

**Figure 5: Blocked ratio of DoEv4 queries performed by VPs in each country/region.**

Figure 5 also reveals two additional phenomena. Firstly, VPs located in China demonstrate better reachability when accessing DoQv4 services. Specifically, only nine DoQv4 domains are inaccessible from VPs in China. However, there are many obstacles for non-Chinese VPs to access DoQv4 domains located in China. Secondly, DoE queries performed by VPs in most countries experience a comparable blocked ratio. This is mainly due to DoE communications between China and most other countries are hampered by considerable obstructions. For better illustration, we plot the blocked ratio of DoE queries across various country/region pairs in Figure 6. The y-axis is the top 20 countries with the most DoEv4

---

[5]Since the blocked ratio of DoE services varies significantly between mainland China and Hong Kong/Macau/Taiwan, "China/Chinese/CN" refers exclusively to mainland China unless otherwise specified in this paper.

domains, and the x-axis is the top 20 countries with the most DoEv4 queries blocked. We can clearly see the two-way blockade of DoE services in China. In addition, VPs in China, Indonesia, and Vietnam exhibit the poorest reachability to DoE services.

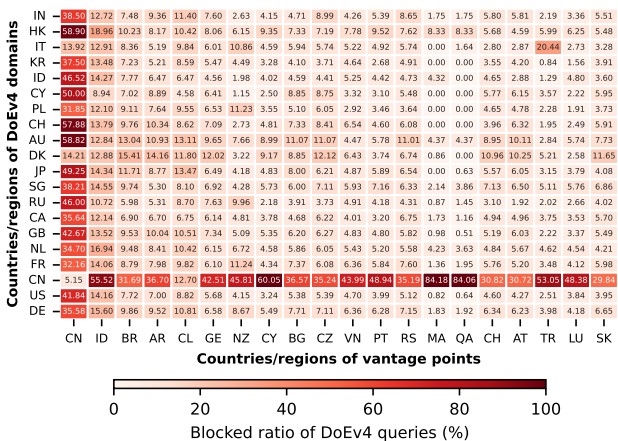

**Figure 6: Blocked ratio of DoE queries between VPs and DoEv4 domains in different country/region pairs.**

Furthermore, we clearly observe DoEv4 service blocking at the AS-level in 12 countries. We define AS-level blocking as the difference in the blocked ratio of DoE queries between ASes in a country exceeding 20%. Remarkably, the AS-level blocking of DoH3v4 services in Russia and Ukraine is particularly prominent. For example, when accessing DoH3v4 services from VPs in AS9009 (Russia) and AS59564 (Ukraine), the blocked ratios are 8.62% and 1.92%, respectively. In contrast, the corresponding ratios for AS50867 (Russia) and AS30860 (Ukraine) are significantly higher at 67.42% and 78.06%.

## 4.3 What blocking types do DoEv4 services suffer?

Digging deeper into blocking types can provide the community with a clearer perception of DoE service accessibility. Figure 7 shows the percentage of blocking types suffered by DoEv4 queries, performed from VPs located in countries with different levels of Internet freedom. We can observe that the reachability of DoE queries performed from VPs in NF countries is poorer, and they experience more frequent blocking in the TCP and QUIC-VN phases. Surprisingly, 62.83% of DoEv4 service are inaccessible due to *Ping blocking*. If other network services share the same IP address as DoE services, this inevitably leads to significant collateral damage. Furthermore, a small fraction of DoE services are affected by *Pre-resolve blocking*. We analyze the blocking types in detail below.

The common *Pre-resolve blocking* behaviors are DNS request timeout (39.77%), and DNS responses present the SERVFAIL code (29.88%) or NXTOMAIN code (13.27%). Furthermore, the majority (40.81%) of *Pre-resolve blocking* are caused by VPs resolving DoE domains located in China. In particular, we observe that some DNS responses are injected with reserved or invalid IP addresses. These behaviors mainly (81.97%) occur when VPs located in China resolve 18 DoEv4 domains associated with a VPN provider [57].

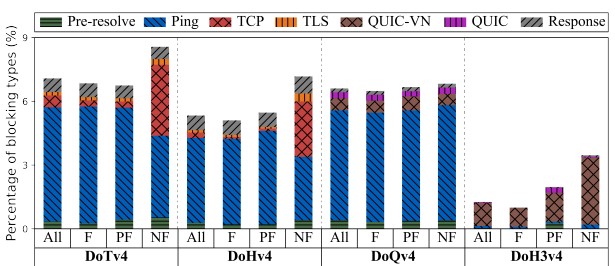

**Figure 7: Statistics on blocking types of DoEv4 queries.**

We observe that VPs in countries with different levels of freedom experience comparable degrees of *Ping blocking*. The primary factor behind this is China's restriction on a considerable amount of VPs. Specifically, 30.97% of ICMP connections between VPs located outside of China and DoE IPv4 addresses within China failed.

The main errors of TCP connection failure are timeout (60.71%) and receipt of RST packets (39.13%). In particular, the DoT service running on the dedicated port 853 suffered from more severe *TCP blocking*. Considering the subsequent TLS handshake, timeout remained the most common error (61.59%), followed by invalid certificates (27.02%). Certificate errors include expiration (79.23%), domain name mismatch (14.36%), and CA untrusted (6.41%). Digging deeper, the primary reason for certificate expiration is the failure of providers to promptly renew certificates for all their servers. In addition, other invalid certificates are mainly due to network middleware injection or intentional behavior by server providers.

Regarding errors during QUIC-VN, 64.88% are connection timeouts and the remainder are connection refused. Furthermore, more DoQ/DoH3 services are blocked during QUIC-VN than subsequent QUIC connections, especially for DoH3 services. This indicates that current blocking strategies for DoQ/DoH3 services generally do not consider the TLS payload (e.g., SNI), but directly block QUIC traffic. For example, all DoQ queries sent to Russia from VPs located in Ukraine are blocked during QUIC-VN.

As for *Response blocking*, most of cases are since DoEv4 responses present the REFUSED code (39.34%) or empty result (33.65%). Furthermore, we find 16 DoEv4 domains respond with non-routable IP addresses. For example, a DoTv4 domain in Russia only returns 0.0.0.0 to some VPs located in the United States.

### 4.4 How is the reachability of DoEv6 services?

In the following, we introduce the reachability of DoEv6 services. During our measurement period, we sent 560K DoEv6 queries to 448 DoEv6 domains from 473 VPs, of which 28K queries were blocked.

Compared to DoEv4, DoEv6 services exhibit better global reachability. Our results show that the blocked ratio of DoEv6 queries performed by VPs in most (88.37%) countries is less than 5%, and China (19.13%) is the only country with a blocked ratio higher than 10%. Surprisingly, VPs located in China are able to access all DoQv6 domains, and only two DoH3v6 domains (dns.google and dns.google.com) are inaccessible. Furthermore, unlike DoEv4, DoEv6 queries are less prone to *Ping blocking* and *TCP blocking*, and failures are more likely to occur during the TLS handshake. Specifically, among blocked DoHv6 queries, 10.61% are *TCP blocking*,

19.21% are *Ping blocking*, and 42.87% are *TLS blocking*. Considering *Pre-resolve blocking*, most VPs that fail to retrieve the correct A record of the DoE domain also encounter difficulties in obtaining the AAAA record. Overall, we recommend that server administrators and client software strengthen support for IPv6 to improve the DoE service reachability.

### 4.5 How many DoE queries are censored?

Censors may block DoE services to ensure their ability to monitor user DNS traffic. In this section, we focus on analyzing whether the motivation behind DoE blocking is censorship. We know that accurately identifying censorship from blocking behavior is difficult. Combined with previous research [17, 41, 55], we list conditions below that strongly indicate that DoE queries are censored.

1) *fake IP address*: DNS responses contain bogon or forged IP addresses. Please refer to section 3.2 for the judgment method.

2) *RST/FIN packet injection*: The TCP reset (RST) or close (FIN) packets are injected into the TCP connection.

3) *self-signed certificate with mismatched domain name*: The DoE server returns a self-signed certificate, and the domain names included in the certificate do not match the DoE domain name.

4) *HTTP(s) blockpage*: The HTTP(s) page scraped from the VP explicitly contains censorship information [46, 49].

5) *HTTP 403 status code*: The HTTP status code of the DoE response returned by the DoH/DoH3 server is 403 (Forbidden).

Our results indicate that 27.18% of blocked DoEv4 queries and 19.73% of blocked DoEv6 queries met at least one of the aforementioned conditions. Since we do not consider complex censorship behaviors, our method may not detect all censored DoE queries. Furthermore, the above six conditions cannot entirely signify censorship, potentially leading to an exaggeration of DoE censorship. However, our findings already demonstrate that censorship has significantly hindered the accessibility of DoE services.

### 4.6 Can clients access blocked DoE domains?

The DoE domain may be hosted on multiple IP addresses and provide various types of DoE services. The strict blocking policy aims to completely prevent clients from accessing DoE services. However, our results suggest that the blocking of some DoE domains is incomplete, which is reflected in two aspects.

The first aspect is that clients can access blocked DoE domains using other IP addresses. For example, VPs in China is unable to establish the DoT session with one IPv4 address (8.8.8.8) of dns.google, but it can receive DoT responses from another IPv4 address and all IPv6 addresses of dns.google. The second aspect is that clients can access blocked DoE domains using other DoE protocols. For example, the DoT/DoH service of dns-family.adguard.com is not accessible in China, while the DoQ/DoH3 service is accessible.

To quantify the incomplete blocking of DoE domains, we filter DoE domains that support multiple IP addresses or DoE protocols. Then, we check the blocked queries between VPs and these DoE domains. The results show that in 59.31% of cases, VPs can use other IP addresses or DoE protocols to access blocked DoE domains. Since our VPs located in China exhibit the worst DoE service reachability, we then take China as an example to analyze the incomplete blocking of DoE services in detail.

| Left ------ Bottom | DoTv4 | DoHv4 | DoQv4 | DoH3v4 | DoTv6 | DoHv6 | DoQv6 | DoH3v6 |
|---|---|---|---|---|---|---|---|---|
| DoTv4 | 46.54 | | | | | | | |
| DoHv4 | 29.96 / **57.66** | 29.27 | | | | | | |
| DoQv4 | 24.06 / **78.19** | 1.65 / **52.52** | 2.88 | | | | | |
| DoH3v4 | 33.33 / **83.33** | 26.67 / **88.33** | **28.57** / 0.00 | 26.67 | | | | |
| DoTv6 | 28.19 / **53.72** | **35.78** / 33.12 | **58.62** / 3.23 | **85.27** / 40.51 | 23.59 | | | |
| DoHv6 | 29.23 / **44.83** | 33.33 / **36.34** | **55.43** / 4.00 | **81.73** / 30.77 | 26.75 / **33.72** | 31.22 | | |
| DoQv6 | 0.00 / **74.19** | 0.00 / **62.72** | 0.00 / **2.94** | 0.00 / **40.24** | 0.00 / **53.67** | 0.00 / **53.67** | 0.00 | |
| DoH3v6 | 15.00 / **80.00** | 11.53 / **86.54** | 0.00 / 0.00 | 11.54 / **30.77** | 15.00 / **85.00** | 11.54 / **81.73** | 0.00 / 0.00 | 11.54 |

**Figure 8: Blocked ratio of VPs located in China to access different DoE service types.**

As shown in Figure 8, we investigate the reachability differences between eight types of DoE services. For example, we select domains that support both DoTv4 and DoHv6, and then calculate their blocked ratios respectively. The bottom number in each white square corresponds to the bottom DoE service type, and the top number corresponds to the left DoE service type. We find that the blocked ratio of DoTv4 services is usually at least 30% higher than other types of DoE services. Furthermore, DoQv6 and DoH3v6 services clearly exhibit better reachability. Hence, we can deduce that China has yet to effectively implement targeted blocking of DoQ/DoH3 protocols, and the block list of IPv6 addresses is not as extensive as that of IPv4 addresses.

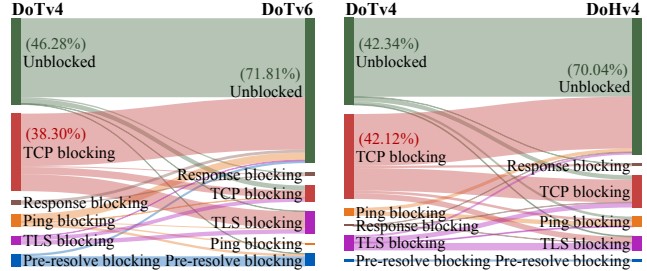

**Figure 9: Blocking changes when VPs located in China access DoTv6/DoHv4 service instead of DoTv4 service.**

Next, we analyze the flow of blocking types when changing DoE service types in China. The left and right subplots in Figure 9 demonstrate the blocking changes for DoTv4 to DoTv6 services (376 domains) and DoTv4 to DoHv4 services (445 domains), respectively. We find that 25.53%/27.70% of blocked DoTv4 domains can still provide DoTv6/DoHv4 services. In addition, many *TCP blocking* strategies specifically target port 853. Therefore, we recommend that new mechanism designs should try to reuse widely used ports and protocols. However, almost all DoTv4 domains that suffer from

*TLS blocking* have no chance of converting to *unblocked*. This indicates that China restricts traffic to some DoE domains based on the SNI field in the TLS handshake.

## 5 DISCUSSIONS

Considering the substantial reliance of emerging privacy protocols upon DoE [21, 51], the enhancement of DoE service accessibility has progressively become a pivotal concern within the community [7, 45, 53]. In the following, we discuss methods for improving the reachability of DoE services that require no extra effort from users.

**Hidden DoE domain name.** Recalling Section 4, the leakage of DoE domain names in DNS queries and TLS/QUIC handshakes may incur targeted blocking. Regarding DNS queries, we recommend that clients embed trusted DoE domains and IP addresses. Although Chrome already does this [9], DoE domain name resolution is also triggered when users access DoE services. Considering TLS/QUIC handshakes, the client can not specify the real DoE domain in the SNI field. To evaluate the effectiveness of this approach, we establish DoE connections from our VPs to each DoE domain without specifying the SNI field. We find that 69.79% of *TLS blocking* and 53.84% of *QUIC blocking* are eliminated.

**Enhance IP address rotation.** Our results indicate that many blocking strategies are based on the IP address of the DoE domain. Therefore, we recommend that providers carefully consider the endpoints hosting their DoE services and ensure rotation of their IP addresses. For instance, opting for CDN platforms or cloud servers with minimal susceptibility to blocking is advisable. In particular, providers can leverage the multi-CDN solution [18] to further improve the global availability of their DoE services. However, from August 7, 2023 to October 9, 2023, we found that the IP addresses associated with 1115 DoEv4 domains and 401 DoEv6 domains remained unchanged.

**Discover the DoE server.** As we mentioned in Section 4.6, the reachability of different DoE service types under the same domain may exhibit huge differences. Nonetheless, clients currently lack a standard method to discover the DoE configuration information of open resolvers. Encouragingly, the Discovery of Designated Resolvers (DDR) [45], proposed by the ADD working group, emerges as a promising solution to this issue. To this end, we use ZMap [15] to collect the IPv4 addresses of open DNS resolvers and check their support for the DDR. We find that 317K DNS resolvers deploy the DDR, of which 243K (76.67%) belong to Google, 39K (12.32%) belong to Cloudflare, and 11K (3.47%) belong to OpenDNS.

## 6 CONCLUSION

Reachability is a basic prerequisite for users to benefit from the DoE mechanism. In this paper, we perform the first comprehensive and large-scale measurement study on the reachability of DoE services. Our findings reveal that DoE services are already widely blocked in some regions. In addition, DoE services over IPv6 exhibit better reachability. A simple yet effective way for clients to access blocked DoE domains is by changing IP addresses or DoE protocols. Our study encourages further exploration by the Internet community into approaches for discovering DoE service information and enhancing the accessibility of DoE services.

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
