# OpenReview forum: "A Worldwide View on the Reachability of Encrypted DNS Services"
_ACM.org/TheWebConf/2024/Conference — TheWebConf24_

### Official Review · Reviewer_bZ8T · 2023-11-07

**Novelty:** 4
**Technical Quality:** 6

**Review:**

This paper presents a large-scale measurement study of the status of DNS-over-Encryption protocols, including the DNS-over-TLS (DoT), DNS-over-HTTPS (DoH), DNS-over-QUIC (DoQ) and DNS-over-HTTP/3 (DoH3). Both IPv4 and IPv6 are tested. The authors discovered over 1K DoE domains, and measured them using 5K VPs for two months. The results show many DoE requests are still blocked. The authors then conducted detailed analysis to identify the type of disruption by the adversaries. In the end, the authors gave some suggestions to enable better DoE resolution.

 This measurement study is comprehensive. I especially like the authors carefully design probing methods to dissect the blocking strategies performed by the adversaries. But I have some concerns.

 1. There have been many measurement studies on DoE services. The authors claim that none of the prior works have both large VP coverage and DoE domain coverage. I don't think this characterization is correct. For example, [35] measured over 1K DoT resolvers with a large number of VPs. I actually think the authors can highlight their contribution in dissecting the blocking strategies.

 2.  There is one related paper not cited [A].

 3. When measuring reachability, [35] and [A] focused on a few public resolvers like Google and Cloudflare. I guess they focus on this small set because they have large user base. Measuring over 1K DoE domains is great, but many of them might not really run DoE services. E.g., they can be mis-configured routers that open to public resolution requests. I suggest authors consider a separate section/subsection describing the results on the known public resolvers.

 4. The authors choose commercial VPN services and EC2 to run their VPs, and they are in data centers or Internet free areas. I wonder why not consider the peer-to-peer proxies used by prior works like [35] and [A], as they can reach censored areas like China.

 5. In Section 3.1, the authors use some keywords and manual rules to identify DoE domains. I wonder the effectiveness of such ad-hoc approach.

 6. In Section 4.5, I would like to see the breakdown numbers of each censor method.

 7. The recommendation of hiding DoE domain in SN is interesting, and it seems to work well. I hope the authors can expand this part.

 [A] Measuring DNS-over-HTTPS Performance Around the World, IMC'21

**Questions:**

1. Can you explain the rationale of using VPN and EC2?

2. Can you have more content regarding known public resolvers?

**Ethics Review Description:**

Internet scanning can cause some ethical issues but the paper discussed them. I think they are addressed well.

**Ethics Review Flag:**

Yes

**Reviewer Confidence:**

4: The reviewer is certain that the evaluation is correct and very familiar with the relevant literature

**Scope:**

4: The work is relevant to the Web and to the track, and is of broad interest to the community

---

### Official Review · Reviewer_mDPp · 2023-11-23

**Novelty:** 3
**Technical Quality:** 5

**Review:**

## Pros
* A comprehensive measurement study of Encrypted DNS services across 102 countries.
* Systematic investigation and classification of the blocking of DoE services.
* Coverage of DoE services over IPv6.
* Interesting findings.

## Cons
* Some findings are already known or reported previously in the literature.
* Few important questions such as how much of the blocking is due to geoblocking policies and how much is due to censorship are not adequately answered.
* Motivation of the study is not appropriately contexualized. The authors mention that studying the reachability of DoE services is essential, but then reachability is also related to coverage of such services, i.e. the extent to which such services are prevalent which is not missing per se, but it is not appropriately contextualized. Would be good to add some numbers of the current prevalence of DoE services and report how much of the total DNS ecosystem has shifted to DoE to understand the significance of the work.

I thank the authors for submitting their work at the Web conference. Overall, I enjoyed reading the paper and appreciate the thoroughness of the measurements. I believe the contribution would be valuable for the community.

1.) There are many interesting findings reported in the paper such as “96/120 blocked DoTv4 domains can provide DoTv6/DoHv4 services in China” That being said, I did felt a lack of explanation and depth in certain parts of the paper. For instance the paper reports that 96/120 v4 domains can provide v6 services. But, don’t talk much about why is that the case or if it’s very easy or difficult for a censor like China given the large impact it can have on their censorship circumvention prevention.

2.) Similarly in Section 3.2, line 306-309, the authors mention that some of the VPNs use DNS hijacking, but don’t talk about why is that the case. A recent paper on analyzing VPNs (VPNalyzer) reports that VPNs sometimes do that to prevent potential DNS leaks outside the VPN tunnel and thus may be better suitable for the kind of evaluation performed in the paper.

3.) In Section 4.5, the authors touch open the censored DoE domains. However, it would have been beneficial to know how much of it was due to censorship and how much of it was geoblocking. The fact that authors did receive 403 forbidden message is indicative of some form of geoblocking and it essential to differentiate the two.

**Questions:**

Can you comment on how easy or difficult it is to incorporate comments 1.) 2.) and 3.) in the above review?

**Reviewer Confidence:**

3: The reviewer is confident but not certain that the evaluation is correct

**Scope:**

4: The work is relevant to the Web and to the track, and is of broad interest to the community

---

### Official Review · Reviewer_QJXG · 2023-11-23

**Novelty:** 5
**Technical Quality:** 5

**Review:**

The paper presents a study of the reachability of encrypted DNS (Dot/DoH/DoQ/DOH3) from a range of vantage points but with a particular focus on China. The results are interesting and relevant for future deployment of these protocols.

There is a discussion of ethics which appears reasonable. It also notes limitations which are significant; specifically that the vantage points in cloud datacenters may not be representative of what a domestic Internet connection might be. However running scans from domestic Internet connections could result in putting people at risk.

I found the term "domain" confusing, since I assumed it meant a group of domain names with a shared subdomain. Eventually I realised that domain is being incorrectly used as a synonym for domain name. Please correct this, or at least note that by domain you mean domain name.

I wondered how comprehensive was the scan. One test would be which proportion of public DoE domain names did it find. Since the list did get augmented after the scan, it looks like a significant proportion of servers are not identified. Is that a fair assumption?

The probing seems to have been done as a sequential process. For example, if DNS lookup fails then the connection is aborted (even though you know the expected IP address). Similarly if the ping test fails, the connection is aborted, even though TCP and QUIC will generally work fine without ICMP. There was a mention of testing connections even if DNS failed, but what about trying the connection even if ping fails?

The text in Figure 6 is too small, and Figure 5 has many countries being invisible. A map is not a very efficient way to show this sort of data. Figure 7 is also very hard to read, and would probably be better as a table.

**Questions:**

- Do I understand correctly that the initial scan for DoE servers was performed only from Hong Kong? If so, could this have introduced bias in the selection of DoE servers identified?
- How many servers would be accessible if the ping test failed but connection was still attempted.

**Reviewer Confidence:**

3: The reviewer is confident but not certain that the evaluation is correct

**Scope:**

4: The work is relevant to the Web and to the track, and is of broad interest to the community

---

### Official Review · Reviewer_NEdk · 2023-11-30

**Novelty:** 3
**Technical Quality:** 5

**Review:**

Summary
=======
DNS over Encryption (DoE) allows Internet users to hide their DNS footprints. However, DoE services are not widely available. This paper conducts a long-term study to collect DoE service providers and analyzes the reachability.


Strengths
=======
+ The experimental studies are supported by data collected over a long period. The data and source code are publicly available which will be useful for future research.
+ The methodology is well-written and provides all necessary information and justification.

Weaknesses
=========
- The actionable insights drawn from the experimental studies are unclear.

**Questions:**

Thank you for your submission. The paper includes valuable datasets and methodologies to analyze the current state of DoE reachability in different countries. Based on the collected data, it reports insights about which countries have weak reachability to DoE due to various reasons such as censorship. The concern the reviewer has is that what are the actionable insights that can be drawn from the results? As those countries intentionally block the access to DoE for governance, should the problem be government policies rather than the technical limitations? The reviewer believes that it is important to clearly identify what the measurement studies can lead to, rather than stating some countries having low reachability.

**Reviewer Confidence:**

1: The reviewer's evaluation is an educated guess

**Scope:**

3: The work is somewhat relevant to the Web and to the track, and is of narrow interest to a sub-community

---

### Official Review · Reviewer_Wo4k · 2023-12-01

**Novelty:** 4
**Technical Quality:** 3

**Review:**

This paper presents a worldwide evaluation of DNS over Encryption (DoE) services, illuminating the extent of their availability in different areas and across multiple protocols. It underscores notable variations in the accessibility of DoE services, pinpointing distinct blocking strategies used across various nations, and emphasizes the importance of improving the worldwide accessibility of these secure DNS services.

Strengths:
- This research provides a global perspective on the accessibility of DNS over Encryption (DoE) services, exploring a diverse array of protocols.
- Conducted over 15 months, this study involved a thorough scan of the Internet, amassing data from 1302 active DoE domains, which encompass both IPv4 and IPv6 addresses, i.e., the authors executed over 10 million DoE queries from 5031 different points of observation in 102 countries.
- The research uncovers seven unique forms of blocking tactics used against DoE services, offering an in-depth understanding of the methods employed to limit access to these services, like Pre-resolve, Ping, and TCP blocking.

Weaknesses:
- The methodology for selecting observation points could lead to biased outcomes. This is because the accessibility of DoE services may differ depending on the geographical area and the network infrastructure in place.
- The findings may not entirely represent the ever-changing nature of online censorship and blocking methods, which tend to adapt quickly to new technological and political shifts.

**Questions:**

Please see my comments above, especially with the weak points.

Minor comment:
- Regarding Figure 6, it is hard to recognize the numbers in the cells.

**Reviewer Confidence:**

2: The reviewer is willing to defend the evaluation, but it is likely that the reviewer did not understand parts of the paper

**Scope:**

4: The work is relevant to the Web and to the track, and is of broad interest to the community

---

### Official Review · Reviewer_FeG9 · 2023-12-01

**Novelty:** 6
**Technical Quality:** 6

**Review:**

The paper  provides a comprehensive analysis of the accessibility and reachability of various types of Domain Name System (DNS) services that utilize encryption (DoE). The study is conducted on a global scale and focuses on different aspects of service reachability, including the extent and types of blocking encountered, variations in accessibility based on geographic location, and the impact of different network conditions and censorship regimes. The methodology involves large-scale measurements using vantage points (VPs) across various countries, assessing the reachability of encrypted DNS services through different protocols. The paper analyzes the types of blocking encountered, such as pre-resolve blocking, TCP and TLS blocking, and response blocking, offering insights into how these services are impacted by network-level censorship and restrictions.

## Strenghts
- The study's global scale provides a broad and detailed perspective on the state of encrypted DNS service reachability worldwide.
- In-depth analysis of blocking and challenges faced by encrypted DNS services

## Areas of Improvement
- More detailed insights into specific regions or countries with unique censorship or network challenges would be a great addition to the paper

**Questions:**

see above

**Reviewer Confidence:**

2: The reviewer is willing to defend the evaluation, but it is likely that the reviewer did not understand parts of the paper

**Scope:**

4: The work is relevant to the Web and to the track, and is of broad interest to the community

---

### Decision · Program_Chairs · 2024-01-22

**Decision:**

Accept

**Comment:**

Our decision is to accept. Please see the AC's review below and improve the work considering that and the reviewers' feedback for cemera-ready submission.

"The paper presents a study of the reachability of encrypted DNS from a range of vantage points. The reviewers appreciated the comprehensiveness of the study and noted the findings to be interesting. However some concerns were raised regarding the novelty -- the authors are recommended to explicitly highlight the novelty in the paper (based on the rebuttal)."